# Methylglyoxal Impairs the Pro-Angiogenic Ability of Mouse Adipose-Derived Stem Cells (mADSCs) via a Senescence-Associated Mechanism

**DOI:** 10.3390/cells12131741

**Published:** 2023-06-28

**Authors:** Alessia Leone, Antonella Nicolò, Immacolata Prevenzano, Federica Zatterale, Michele Longo, Antonella Desiderio, Rosa Spinelli, Michele Campitelli, Domenico Conza, Gregory Alexander Raciti, Francesco Beguinot, Cecilia Nigro, Claudia Miele

**Affiliations:** URT Genomics of Diabetes, Institute of Experimental Endocrinology and Oncology, National Research Council & Department of Translational Medical Sciences, Federico II University of Naples, 80131 Naples, Italy; a.leone@ieos.cnr.it (A.L.); antonellan43@gmail.com (A.N.); immacolata.prevenzano@unina.it (I.P.);

**Keywords:** adipose-derived stem cells, advanced glycation end-products, angiogenesis, cell migration, dicarbonyl stress, endothelial cells, methylglyoxal, senescence

## Abstract

Adipose-derived stem cells (ADSCs) play a crucial role in angiogenesis and repair of damaged tissues. However, in pathological conditions including diabetes, ADSC function is compromised. This work aims at evaluating the effect of Methylglyoxal (MGO), a product of chronic hyperglycemia, on mouse ADSCs’ (mADSCs) pro-angiogenic function and the molecular mediators involved. The mADSCs were isolated from C57bl6 mice. MGO-adducts and p-p38 MAPK protein levels were evaluated by Western Blot. Human retinal endothelial cell (hREC) migration was analyzed by transwell assays. Gene expression was measured by qRT-PCR, and SA-βGal activity by cytofluorimetry. Soluble factor release was evaluated by multiplex assay. MGO treatment does not impair mADSC viability and induces MGO-adduct accumulation. hREC migration is reduced in response to both MGO-treated mADSCs and conditioned media from MGO-treated mADSCs, compared to untreated cells. This is associated with an increase of SA-βGal activity, SASP factor release and p53 and p21 expression, together with a VEGF- and PDGF-reduced release from MGO-treated mADSCs and a reduced p38-MAPK activation in hRECs. The MGO-induced impairment of mADSC function is reverted by senolytics. In conclusion, MGO impairs mADSCs’ pro-angiogenic function through the induction of a senescent phenotype, associated with the reduced secretion of growth factors crucial for hREC migration.

## 1. Introduction

The regenerative and immunomodulatory properties of mesenchymal stem cells (MSCs) have attracted great attention worldwide, because of their potential for cell-based therapy [1]. Among them, adipose-derived stem cells (ADSCs) are harvested from adipose tissue and are considered a promising tool for their use in regenerative medicine, including the treatment of degenerative, inflammatory and autoimmune diseases [2]. Indeed, strong evidence supports the effectiveness of ADSCs in the treatment of multiple sclerosis, rheumatoid arthritis, osteoarthritis, diabetes mellitus (DM), dyslipidemia and cardiovascular disease, as well as skin aging and wound healing [2]. The interest in ADSCs’ use as a therapeutic approach is due to their ability to enhance angiogenesis and accelerate tissue healing [3]. Indeed, angiogenesis is necessary for tissue repair; an adequate vascular network is crucial for guaranteeing blood and growth factor delivery to injured tissues as well.

ADSCs stimulate angiogenesis in several ways. Their ability to differentiate into endothelial cells (ECs) and smooth muscle cells has been described, indicating a possibly direct role in vascularization in vivo [4]. They can also differentiate towards pericyte-like cells, which play an important role in the stabilization of the vessel wall and the prevention of vascular leakage [5]. Beyond these roles, ADSCs can indirectly sustain angiogenesis in a paracrine manner. Indeed, they produce and release a broad spectrum of biologically active factors including growth factors, cytokines, chemokines, cell adhesion molecules, extracellular vesicles (EVs) and RNA, factors that support cell regeneration, proliferation, differentiation and migration [2]. However, in some pathological conditions, such as in the case of chronic hyperglycemia and metabolic disorders typical of DM, the functionality of ADSCs is compromised. Indeed, ADSCs isolated from patients with coronary heart disease and DM show reduced angiogenic activity due to an imbalance of pro- and anti-angiogenic growth factors secreted by ADSCs [6]. Furthermore, a study by Inoue et al. demonstrates that diabetic ADSCs (DM-ADSCs) show a reduced cellular proliferation and VEGF secretion associated with a reduced expression of the stemness-related gene SOX2. This leads to an impairment of in vivo angiogenic ability when DM-ADSCs are administrated in a mouse model of hindlimb ischemia [7]. Similarly, ADSCs isolated from diabetic mice show impaired functionality in promoting wound healing [8]. Therefore, the impairment of ADSC function represents the main limitation for the use of autologous ADSCs in regenerative medicine.

Hyperglycemia, oxidative stress and altered immune reactions are well-known features of a diabetic microenvironment, which are associated with a change in ADSC properties and functions, including ADSC senescence [1].

Methylglyoxal (MGO) is a highly reactive dicarbonyl primarily formed as a byproduct of glycolysis [9]. It accumulates in chronic hyperglycemia, inducing dicarbonyl stress and the increased formation of advanced glycation end-products (AGEs) [10]. MGO induces cellular dysfunction through several mechanisms including cellular senescence [11]. We have recently demonstrated that MGO accumulation induces a β-cell senescence-associated pro-inflammatory phenotype that contributes to the establishment of an early phenotype typical of type 2 diabetes mellitus (T2DM) progression [12].

It is known that MGO accumulation exerts a harmful effect on vasculature and is involved in the development of diabetic complications, causing damaging effects both on macro- and micro-vasculature [13]. In particular, we have previously demonstrated that MGO accumulation impairs the angiogenesis of ECs’ knock-down of the main enzyme of MGO detoxification, the Glyoxalase 1 (Glo1) [14].

It has been recently shown that MGO is able to inhibit the differentiation of human MSC-derived osteoblasts in vitro [15]. Conversely, the overexpression of Glo1 can restore the impaired cell viability, migration, differentiation and pro-angiogenic ability of ADSCs both in vitro and in vivo, in a diabetic ischemic model [16]. However, no information is available about the direct effect of MGO on the pro-angiogenic function of ADSCs and the potential mechanisms involved.

Therefore, the aim of this study is to investigate the impact of MGO accumulation on the ability of ADSCs to sustain the angiogenesis of ECs.

We here provide novel information on the molecular aspects involved in the impaired migration of human endothelial retinal cells (hRECs) in response to MGO-treated mouse ADSCs (mADSCs), highlighting a decrease in vascular endothelial growth factor (VEGF) and platelet-derived growth factor (PDGF) secretion by mADSCs. This is associated with a senescence phenotype of these cells and an impaired activation of the cell migration mediator p38 MAPK in hRECs, which is rescued by senolytic treatment.

## 2. Materials and Methods

### 2.1. Reagents

Media, sera and antibiotics for cell culture were provided by Innoprot (Derio, Bizkaia, Spain) and Gibco (Waltham, MA, USA). MGO (40% in water) was from Sigma-Aldrich (St. Louis, MO, USA). Protein electrophoresis and western blot reagents were from Bio-Rad (Richmond, VA, USA) and ECL reagents from Pierce (Rockford, IL, USA). The antibodies used for western blot are anti-MGO (Abcam, Trumpington, Cambridge, UK) and anti-14.3.3 (Santa Cruz Biotechnology, Dallas, TX, USA), anti-vinculin (Santa Cruz Biotechnology, Santa Cruz, CA, USA) and phospho-p38 MAPK (Thr180/Tyr182) (Cell Signaling, Technology, Danvers, MA, USA). CD29-APC, CD44-PE, CD45-FITC, CD31-APC, CD90.2-PE and Sca-1-APC antibodies were provided by Miltenyi Biotec (Auburn, CA, USA). TRIzol and SuperScript III were from Invitrogen (Carlsbad, CA, USA). SYBR Green Supermix was from Bio-Rad (Hercules, CA, USA). All other chemicals were from Sigma-Aldrich (St. Louis, MO, USA).

### 2.2. mADSCs Isolation and Culture

ADSCs were isolated from subcutaneous adipose tissue biopsies (1 g) of 4-month-old C57bl6 mice. Tissue collection from mice was approved by the local ethics committee of the Ministry of Health (approval n° 252/218-PR). Under a laminar flow hood, the biopsies were washed with DMEM/F12 (Dulbecco’s Modified Eagle Medium/Nutrient Mixture F-12) and minced mechanically with sterile scissors. The finely shredded tissues were processed with the “Adipose tissue Dissociation kit” (Miltenyi Biotec, Auburn, CA, USA) according to the manufacturer’s instructions. The digested tissues were then filtered using 100 μm nylon mesh and centrifuged at 1500 rpm for 15 min. After centrifugation, the pellet, consisting of the stromal vascular fraction (SVF) containing the mADSCs, was resuspended in the DMEM/F12 supplemented with 10% fetal bovine serum (FBS), Pennicillin 100 (U/mL), Streptomycin (50 μg/mL) and Amphotericin B (1.5 μg/mL) and plated in a T25 flask. Cells were cultured for 5 passages before proceeding with mADSCs characterization. Experiments were performed in at least 3 different isolations of mADSCs from C57bl6 mice.

### 2.3. mADSCs Characterization

(a)Immunophenotype

Once confluence was reached, the cells were collected and the pellet was washed with 1× PBS. After centrifuging at 300× *g* for 10 min and removing the supernatant, the pellet was resuspended in 400 μL of cold buffer (PBS pH 7.2, BSA 0.5%, EDTA 2 mM). Then, the cells were divided into 4 aliquots: (i) stained with APC-labeled anti-CD29 (Miltenyi Biotec 130-119-166), FITC-labeled anti-CD45 (Miltenyi Biotec 130-110-796) and PE-labeled anti-CD90.2 (Miltenyi Biotec 130-120-897) in 100 μL of PBS; (ii) PE-labeled anti-CD44 and APC-labeled anti-CD31 antibodies in 100 μL of PBS; (iii) APC-labeled anti-Sca-1 (Miltenyi Biotec 130-123-848) in 100 μL of PBS; or (iv) used as negative control. Samples were placed for 10 min at 4 °C in the dark, washed with 1 mL of PBS and centrifuged at 300× *g* for 10 min at 4 °C. Subsequently, samples were resuspended in 300 μL of PBS with the BD LSRFortessa FACS (BD Biosciences, San Jose, CA, USA) and analyzed using the BD FACS Diva software v6.2. For each experiment, 10,000 cells were counted.

(b)Adipogenic differentiation

2.5 × 10^5^ mADSCs were plated in MW6 and cultured in DMEM/F12 10% FBS until 80% confluence was reached (day-2). After two days (day 0), they were induced to differentiate in DMEM/F12 (1:1) 3% FBS with the addition of a differentiation mix consisting of 850 nM insulin, 0.5 mM IBMX (3-isobutyl-1-methylxanthines), 10 μM dexamethasone and 10 μM rosiglitazone. After three days (day 3), the medium was replaced with DMEM/F12 (1:1) 10% FBS supplemented with 850 nM Insulin, 1 μM dexamethasone and 1 μM rosiglitazone, and changed every three days until the end of differentiation protocol (21 days).

(c)Oil Red-O staining

After 21 days of differentiation protocol, the cells were stained with Oil Red-O. After removing the culture medium, the adipocytes were washed twice with PBS. Subsequently, cells were fixed with 4% formaldehyde and incubated at room temperature (RT) for 5 min. After two washes with PBS, the plates were washed quickly with 60% isopropyl alcohol and then incubated for 30 min at RT with Oil Red-O. Cells were then washed twice with PBS and observed under a light microscope. Subsequently, samples were eluted in 60% isopropanol and the color intensity was quantized using the BECKMAN model spectrophotometer (DU-730), evaluating the absorbance at the wavelength of 490 nm.

(d)Osteogenic differentiation

2.5 × 10^5^ mADSCs were plated in MW6 and cultured in DMEM/F12 10% FBS. Once confluence was reached, they were induced to differentiate as previously described [17]. The medium was changed every 48 h until the end of differentiation protocol (21 days).

(e)Alizarin Red S staining

After 21 days of differentiation protocol, the cells were stained with Alizarin Red S (ARS) as previously reported [17]. Briefly, cells were fixed with 4% formaldehyde and incubated at room temperature (RT) for 15 min, washed with bidistillated water and stained with ARS pH 4.1 for 20 min at RT with gentle shaking. The stained monolayer was observed under a light microscope. For quantification, 800-μL 10% acetic acid was added to each well. Cell monolayers were scraped and transferred to a 1.5-mL microcentrifuge tube. After vortexing for 30 s, tubes were heated at 85 °C for 10 min, transferred on ice for 5 min and centrifuged at 20,000× *g* for 15 min. 500 μL of the supernatant was removed, transferred to a new tube and 200 μL of 10% ammonium hydroxide was added to neutralize the acid. ARS staining was assessed by optical density determination at 490 nm using a microplate reader.

### 2.4. Cell Culture Procedure

hRECs were purchased by Innoprot (Derio, Bizkaia, Spain) and cultured in T75 flasks previously coated with fibronectin (Innoprot, Derio, Bizkaia, Spain) using EC Medium consisting of 500 mL of basal medium supplemented with 25 mL of FBS, 5 mL of EC growth supplement (ECGS) and 5 mL of a penicillin/streptomycin solution (P/S solution).

mADSCs were cultured in T25 flask and grown in DMEM/F12 with 10% FBS, Pennicillin 100 (U/mL) and Streptomycin (50 μg/mL).

Cell cultures were maintained at 37 °C in a humidified 5% (*v*/*v*) CO_2_ incubator. Where indicated, cells were treated with 100 µM MGO (Sigma-Aldrich, St. Louis, MO, USA) for 16 h and treated or not treated with dasatinib (0.25 µM) and quercetin (10 µM), in combination, for 24 h (Sigma-Aldrich, St. Louis, MO, USA).

### 2.5. Cell Viability

10^4^ mADSCs *per* well were seeded in MW96 and grown in the standard culture medium described above. Subsequently, they were treated or not treated with increasing concentrations (25 μM, 50 μM, 100 μM, 200 μM, 300 μM, 400 μM, 500 μM) of MGO for 16 h. The viability of the mADSCs was then assessed by using the “MTT Cell Viability Assay” kit (Biotium, Fremont, CA, USA) according to the manufacturer’s instructions. The sample was read using a spectrophotometer (Infinite 200, Tecan, Mannedorf, Switzerland) at 595 nm.

### 2.6. Population Doubling Time (PDT)

3.0 × 10^5^ mADSCs at passage 2 (p2) were plated in T25 and cultured as described above. Once 90% confluence was reached, cells were detached and subcultured again up to p11. At each subculture, numbers of harvested cells were counted using Automated Cell Counter TC20 (BioRad, Hercules, CA, USA). PDT and accumulated cell number were deduced by the following equation: T × log (2)/log (q2) − log (q1); T, cell culture time; q1, initial number of cells; q2, final number of cells [18].

### 2.7. Western Blot Analysis

Total protein lysates were obtained and separated by SDS-PAGE as previously described [14]. Upon incubation with primary and secondary antibodies (for full list see above), immunoreactive bands were detected by chemiluminescence, and densitometric analysis was performed using ImageJ software 1.46r.

### 2.8. RNA Isolation, Reverse Transcription and Real-Time PCR

Total RNA was isolated from mADSCs using TRIzol reagent according to the manufacturer’s protocol. After quantification with NanoDrop 2000 spectrophotometer (Thermo Scientific, Waltham, MA, USA), 1 μg of total RNA was reverse-transcribed using SuperScript III according to the manufacturer’s instructions. The differential expression of genes was analyzed by quantitative real-time PCR, as previously described [14], and quantified as relative expression units using the comparative 2^−ΔΔCt^ method. Cyclophilin A was used as housekeeping gene. Specific primers used for amplification were purchased from Sigma-Aldrich (St. Louis, MO, USA) and listed in Table 1 below:

### 2.9. Migration and Co-Culture Assays

hREC migration was evaluated through a 2D co-cultures system using MW24 “Transwell^®^ Permeable Supports” inserts (BD Falcon, Franklin Lakes, NJ, USA) containing 8 μm pore polyethylene terephthalate (PET) membrane. mADSCs were seeded in MW24, cultured with DMEM/F12 with 10% FBS and treated or not treated with 100 μΜ of MGO for 16 h. DMEM/F12 was then replaced with Opti-MEM basal medium (Gibco, Waltham, MA, USA) and 5 × 10^4^ hRECs per well were seeded in EC basal medium on the upper side of 8.0 μm transwell inserts placed inside the mADSC-containing wells. As negative control, hRECs were seeded on the upper side of the insert placed in a well containing Opti-MEM in the absence of mADSCs.

For the migration assay in response to conditioned media (CM), mADSCs were seeded in MW24 and treated or not treated with 100 μΜ of MGO for 16 h. Where indicated, following MGO treatment, mADSCs were treated with dasatinib 0.25 μΜ and quercetin 10 μΜ, in combination. The medium was then removed and replaced with Opti-MEM for 24 h to obtain the CM. After 24 h, the CM from mADSCs was collected, centrifuged and placed in MW24 for the hREC migration assay. Subsequently, hRECs were seeded as reported above. Not conditioned Opti-MEM (basal medium) was used as negative control of hRECs migration.

Following 24-h incubation, the top of the insert was cleared of cells with a cotton swab. The top and the bottom of the insert were washed 3 times with 1× PBS. Inserts were incubated for 30 min at room temperature with 11% glutaraldehyde to fix the cells migrated to the lower side of the insert, and then washed 3 times with 1× PBS. Cells were then stained with Crystal Violet and washed with 1× PBS. Cells from 4 representative fields from each insert were counted [14].

### 2.10. Detection of Senescence-Associated Beta-Galactosidase (SA-βgal) Activity

SA-βgal activity has been evaluated in vitro by a fluorescence-based assay using flow cytometry, as described in [19]. Cell suspension was run in a BD FACS and the acquisition and analysis were performed by BD FACSDiva Software v6.2 (BD Bioscience, San Jose, CA, USA).

### 2.11. Multiplex Immunoassay

CM from mADSCs were obtained as described above. The levels of secreted mouse cytochines, chemokines and growth factors were measured by the use of a 23-plex (#M60009RDPD) and 9-plex (#MD000000EL) ELISA multiplex assay (Bio-Plex ProTM, Bio-Rad, Hercules, CA, USA) according to manufacturer’s protocol. Data were acquired using a Bio-Plex 200 system equipped with Bio-Plex Manager software v5.0 (BioRad, Hercules, CA, USA). The standard curve optimization and the calculation of analyte concentrations were performed by using the Bio-Plex Manager software. Unconditioned Opti-MEM (basal medium) was used as blank and reading values were subtracted to sample values.

### 2.12. Statistical Procedures

Data are expressed as means ± SEM of at least three independent experiments. Comparisons between groups were performed using Student’s *t*-test. *p*-values of less than or equal to 0.05 were considered statistically significant.

## 3. Results

### 3.1. Characterization of Isolated mADSCs

mADSCs were isolated starting from 1 gr of mouse subcutaneous adipose tissue. The total amount of cells obtained after tissue digestion (1.42 ± 0.52 × 10^7^) were cultured as described in the Materials and Methods section for mADSC selection (Figure 1a). Cell number and population doubling time (PDT) are indicated in Figure 1b,c.

mADSCs were characterized by analyzing the expression of surface markers by FACS and their ability to differentiate in adipocytes and osteoblast by Oil Red-O and Alizarin Red-S staining, respectively. As expected, mADSCs express the surface markers typical of stem cells. In detail, mADSCs show more than 95% positivity for CD44, CD29, CD90.2 and Sca-1 (Figure 1d–g) and less than 2% positivity for both CD45 and CD31 (Figure 1h,i). Moreover, mADSCs are able to differentiate in adipocytes in response to specific conditions, as indicated by the increase in *Pparɣ2*, *Glut4*, *Ap2* and *AdipoQ* gene expression, and by a 5-fold increase in Oil Red-O staining (Appendix A). Furthermore, the increased expression of *Runx2*, *Msx2*, *Ocn* and *Ocp* genes and a 12-fold increase in Alizarin Red-S staining indicate that mADSCs are also able to differentiate in osteoblast (Appendix A).

Altogether, these data show that the used isolation protocol is a valid method for obtaining mADSCs and is useful for carrying out the planned studies.

### 3.2. MGO Treatment Does Not Impair mADSC Viability and Induces an Accumulation of MGO-Adducts

In order to analyze the effect of MGO on mADSCs, dose–response experiments were performed to identify the concentration of MGO that was not toxic for these cells. As shown in Figure 2a, the treatment up to 100 µM of MGO for 16 h does not impair cell viability that is conversely reduced starting from 200µM of MGO concentration. Moreover, mADSCs treated with MGO 100 µM show a 1.7-fold increase in MGO-adduct accumulation compared to untreated control cells (Figure 2b,c).

For these reasons, this experimental condition has been chosen for the accomplishment of our study.

### 3.3. MGO Treatment Impairs the Pro-Angiogenic Ability of mADSCs

To test the pro-angiogenic ability of mADSCs, these cells were co-cultured with hRECs and the migration of hRECs was analyzed following 24 h of co-culture by evaluating the number of cells migrated through a porous membrane in the presence or absence of mADSCs.

The representative images in Figure 3a show that the presence of mADSC is able to induce a massive migration of hRECs. However, the treatment of mADSCs with MGO impairs this ability. Indeed, hREC migration is reduced by 46% when co-cultured with MGO-treated mADSCs (Figure 3b).

### 3.4. MGO Treatment Induces Senescence in mADSCs

It is known that ADSCs are susceptible to hyperglycemia-induced oxidative stress-mediated senescence with activation of p53 and growth arrest [20,21,22]. Furthermore, increased levels of glyoxal (GO) and MGO, observed in diabetes and aging, can induce cellular senescence in ECs [11].

In order to demonstrate the hypothesis that the pro-angiogenic function of mADSCs might be compromised by an MGO-induced senescence phenotype, the expression of specific markers has been evaluated in mADSCs. As shown in Figure 4, both *Trp53* (Figure 4a) and *Cdkn1a* (Figure 4b) mRNA levels (the genes encoding for p53 and p21 proteins, respectively) are increased by 1.7-fold in mADSCs treated with MGO, compared to untreated control cells. Moreover, the activity of SA-β-galactosidase results in a 2.2-fold increase in MGO-treated mADSCs compared to untreated control cells (Figure 4c).

Subsequently, to investigate whether the treatment with MGO induced the release of senescence-associated secretory phenotype (SASP) factors, a panel of mouse cytokine, chemokine and growth factors was analyzed in the CM obtained from mADSCs treated or not treated with MGO and maintained in culture for 24 h. As indicated in Table 2, the SASP factors interleukin-6 (IL-6), monocytes chemotactic protein 1 (MCP1) and interleukin-12 (IL-12) are released at higher levels in the CM by MGO-treated mADSCs compared to controls.

Therefore, mADSCs show molecular signs of senescence following the exposure to MGO.

### 3.5. MGO Treatment Impairs the Soluble Factor Dependent Pro-Angiogenic Ability of mADSCs

To test whether the MGO-induced impairment of mADSCs pro-angiogenic activity was mediated by an altered release of soluble factors, hRECs migration was analyzed in response to CM from mADSCs.

CM from untreated control mADSCs induces a substantial increase in hREC migration compared to the basal medium. Interestingly, hREC migration is reduced by 34% in response to CM from MGO-treated mADSCs compared to untreated mADSCs (Figure 5).

Protein levels of nine factors involved in the angiogenic process were analyzed by an ELISA multiplex assay in the CM from mADSCs. Five out of nine factors were detected in the CM and among these, both VEGF and PDGF-BB levels are reduced in the CM from mADSCs previously exposed to MGO, compared to untreated control mADSCs (Table 3).

### 3.6. p38 MAPK Activation Is Reduced by CM from MGO-Treated mADSCs in hRECs

In light of the impaired release of VEGF and PDGF-BB by MGO-treated mADSCs, we looked at the activation of a key factor involved in the induction of cellular migration, p38 MAPK, as a potential mediator downstream both VEGF and PDGF pathways.

The activation of p38 MAPK has been evaluated on protein lysates of hRECs exposed to CM derived from mADSCs treated or not treated with MGO. As shown in Figure 6, phopsho-p38 MAPK levels are ~4.7-fold increased in hRECs exposed to CM from control mADSCs. Interestingly, the phopsho-p38 MAPK levels are reduced by 50% in hRECs exposed to CM from MGO-treated mADSCs compared to hRECs exposed to CM from control mADSCs.

### 3.7. Senolytic Treatment of mADSCs Exposed to MGO Restores Their Ability to Reduce Migration of hRECs

To test whether the senescence of mADSCs was crucial for the detrimental effect of MGO on mADSCs’ pro-angiogenic function, the latter were treated with 0.25 µM dasatinib and 10 µM quercetin, in combination, for 24 h before media conditioning.

Treatment of mADSCs with senolytics is able to significantly increase the levels of both VEGF and PDGF in CM from MGO-treated mADSCs (Table 4).

Furthermore, the activation of p38-MAPK, found to be impaired by the hRECs exposure to the CM from MGO-treated mADSCs, is rescued in hRECs in response to the CM from mADSCs treated with dasatinib and quercetin following exposure to MGO (Figure 7a,b). Interestingly, senolytic treatment is also able to rescue the downstream migration of hRECs in response to CM from mADSCs exposed to MGO (Figure 7c,d).

## 4. Discussion

Given the ease of collection through minimally invasive procedures, ADSCs are considered as a medically vital resource for modern regenerative medicine [4]. Indeed, to date, a great number of clinical trials involve the use of ADSCs for the treatment of tissues damaged by ischemic injury [4]. The main role played by ADSCs in the regeneration of damaged tissues is linked to their pro-angiogenic activity, as these cells activate and recruit endogenous cells involved in angiogenesis and stabilize newly formed vessels [23]. However, pathological conditions like DM and the related hyperglycemia compromise ADSC functions, therefore limiting their use in regenerative medicine [6]. The use of autologous ADSCs gives the advantage of low immunogenicity and bypasses any ethical and legal concerns compared to other stem cell types (i.e., embryonic stem cells or induced pluripotent stem cells) [24]. Therefore, there is an urgent need to unveil the molecular alterations induced by glucotoxicity to be targeted for the optimization of diabetic patient-derived ADSC function.

In this study, we aim at evaluating if the accumulation of MGO, a byproduct of chronic hyperglycemia known to be at high levels in diabetic patients [25], impairs the pro-angiogenic ability of ADSCs isolated from the subcutaneous adipose tissue of C57bl6 mice (mADSCs), in order to identify the molecular mechanisms involved.

The International Society for Cell Therapy (ISCT) guidelines establish that adhesion growth, the presence or absence of specific surface antigens and the ability to differentiate in other cell lines represent the three minimum criteria for the identification of ADSCs [26]. The immunophenotype specific to ADSCs from mice has been delineated by several authors in the literature [27,28,29]. Accordingly, we verified that isolated mADSCs show more than 95% positivity for the stem cell markers CD29, CD44, CD90.2 and Sca-1, and less than 2% positivity for CD45 and CD31 markers, typical of other cellular components of SVF. Moreover, their differentiation ability in adipocytes and osteoblasts made us confident that the protocol used for mADSC isolation was valid for the accomplishment of our study.

The accumulation of the highly reactive dicarbonyl MGO is associated with the development of microvascular complications in subjects suffering from DM and a correlation between serum MGO-H1 levels and the progression of diabetic retinopathy (DR) has been found in diabetic subjects [30]. MGO exerts its action by mainly interacting with the arginine and serine residues of proteins. They can be quantified by immunoblotting [31]. In particular, the modification of arginine by MGO generates hydroimidazolones, of which MGO-H1 represents the main AGE deriving from MGO [13]. Proteins susceptible to MGO modification are part of the so-called “dicarbonyl proteome”, and many of them have been identified in different biological samples [32,33]. Here, we do not pinpoint specific proteins whose function is modified by MGO-conjugation but verify that our experimental conditions foster the accumulation of MGO-adducts, which happens in diabetes and aging. Moreover, this treatment does not impair mADSC viability. For this reason, the exposure of mADSCs to 100 µM MGO was chosen as the experimental condition for studying the effect of MGO on the pro-angiogenic ability of mADSCs.

The functionality of ECs is crucial for physiological angiogenesis. Following pro-angiogenic stimuli, ECs leave their state of quiescence to start proliferating, migrating and invading the extracellular matrix (ECM) to form new blood vessels [34]. MGO is known to have a detrimental effect on endothelial function [35]. We have previously demonstrated that a non-toxic treatment with MGO for 16 h induces endothelial insulin resistance both in vitro and in vivo [36]. Furthermore, the prolonged exposure to MGO induces alterations in the wound repair process and vascular damage in rodent models [37]. We also showed that high levels of MGO impair the angiogenic capacity of murine aortic ECs knocked down for Glo1 [14].

Several studies demonstrate that ADSCs interact with ECs and promote their migration and recruitment for endothelialization [38]. Lupo et al. have recently demonstrated that pericyte-like differentiated ADSCs preserve the blood-retinal barrier in an in vitro model of DR playing a positive effect on hRECs damaged by high glucose [39]. Moreover, Peng et al. reported that the overexpression of the MGO-metabolizing enzyme Glo1, in ADSCs isolated from diabetic mice, is able to rescue the impaired ability of these cells to induce angiogenesis in human umbilical vein endothelial cells (HUVECs) [16]. Similarly, we use here a cellular system consisting of mouse-isolated ADSCs and human ECs in co-culture and confirm the ability of mADSCs to stimulate the migration of hRECs. Interestingly, the treatment of mADSCs with MGO impairs their effect on hREC migration, demonstrating for the first time a direct effect of MGO on mADSCs’ pro-angiogenic function.

In recent years, the ADSC secretome has been used in clinical trials as a safer and more effective alternative to whole-cell administration [2]. Indeed, one of the mechanisms by which ADSCs carry out their pro-angiogenic action is through the release of soluble factors [23]. The role of ADSC-CM in promoting wound healing, migration and angiogenesis is sustained by many studies [40,41]. Conversely, it has been shown that ADSCs from diabetic mice release lower levels of pro-angiogenic factors (i.e., HGF, VEGF and IGF-1) decreasing both proliferation and migration of fibroblasts and keratinocytes in vitro [42]. Moreover, Inoue et al. reported that ADSCs derived from T2DM patients show a reduced secretion of VEGF, which are associated with an impaired in vivo angiogenic capacity in xenograft experiments using a mouse model of hindlimb ischemia [7].

Consistently, our results demonstrate that CM obtained from mADSCs strongly induce hREC migration, which is reduced when hRECs are exposed to CM from MGO-treated mADSCs. The impaired pro-angiogenic ability of mADSCs is associated with the modulation of soluble factors released by mADSCs in the extracellular environment. Indeed, the analysis of soluble factors secreted by mADSCs highlighted the reduced release of VEGF and PDGF in the CM from MGO-treated mADSCs, compared to untreated cells.

Recent evidence sustains the contribution given by the activation of senescence pathways to the detrimental effect of MGO on cellular function. Indeed, increased levels of dicarbonyls, observed in diabetes and aging, induce cellular senescence in ECs [11]. We recently verified the crucial role of a premature β-cell senescent phenotype in vivo in the alteration of glucose homeostasis induced by MGO accumulation [12]. Senescent ADSCs lose their angiogenic potential and the transplantation of them causes dysfunction in mice [22,43,44,45,46]. Whilst a growing body of literature claims that both senescence and the SASP are sensitive to metabolic states (i.e., hyperglycemia), a degenerative feedback loop can be established by cellular senescence, which in turn promotes diabetes and degenerative complications [47]. Mechanistic details by which hyperglycemia can drive senescence are still needed.

Oxidative stress induces a premature senescence of MSCs, known as oncogene-induced or stress-induced senescence, which anticipates the replicative senescence that usually occurs following passaging in culture [45]. The senescent MSCs are characterized by an impairment of their function and reduced therapeutic effects. Indeed, CM obtained from senescent AT-MSCs attenuate their angiogenic potential [44], and targeting senescence is able to improve the angiogenic ability of ADSCs in subjects with preeclampsia [43]. In light of this evidence, to further clarify the molecular mechanisms underlying the MGO-induced impaired functionality of mADSCs, we tested whether cellular senescence was involved in this effect.

In our experimental model, the mRNA levels of p53 and p21 are increased in MGO-treated mADSCs compared to untreated control cells. Moreover, a higher activity of SA-βGal is also associated with the increased release of the SASP factors IL-6, MCP1 and IL-12 in MGO-treated mADSCs. Taken together, these data suggest that the impairment of mADSCs’ pro-angiogenic ability induced by MGO could be related to the activation of senescence pathways leading to the reduced release of the growth factors VEGF and PDGF.

As common signaling factors downstream VEGF and PDGF receptors, we analyzed the activation of p38 MAPK, Akt and ERK1/2, which play a pivotal role in cell migration, survival and proliferation [48,49]. While our preliminary data indicate that the activation of both Akt and ERK1/2 is not modified in hRECs in response to CM from MGO-treated mADSCs, we found that MGO treatment of mADSCs impairs p38-MAPK phosphorylation induced by CM from untreated control mADSCs in hRECs, confirming its involvement in the damaging effect played by MGO. Consistent with our results, p38-MAPK activation leads to actin remodeling, angiogenesis and DNA damage response in ECs [50,51]. Moreover, CM from MSCs overexpressing VEGF promotes islet microvascular endothelial cell migration through p38/MAPK pathway activation [52].

Interestingly, the treatment of mADSCs with senolytics, quercetin and dasatinib in combination, is able to increase the release of VEGF and PDGF from MGO-treated mADSCs. This leads to a rescue of the hREC migration, as well as the p38-MAPK activation, in response to CM from MGO-treated mADSCs. Obtained data prove that the clearance of senescent cells is useful to restore the molecular pathways compromised by MGO, which are crucial for the pro-angiogenic effect of mADSCs on hRECs.

## 5. Conclusions

Here, we demonstrate that MGO accumulation impairs the pro-angiogenic function of mADSCs, shown by the impaired ability of mADSCs to release both VEGF and PDGF and induce hREC migration. We point at the mADSC senescence as the underlying mechanism that contributes, at least in part, to the reduced activation of p38-MAPK, a well-known mediator of cellular migration.

Together, the data obtained in this study provide novel information about the direct harmful effect of MGO on mADSC functionality, paving the way for the optimization of therapeutic strategies involving the use of autologous ADSCs for the treatment of diabetic-associated vascular defects.

## Figures and Tables

**Figure 1 cells-12-01741-f001:**
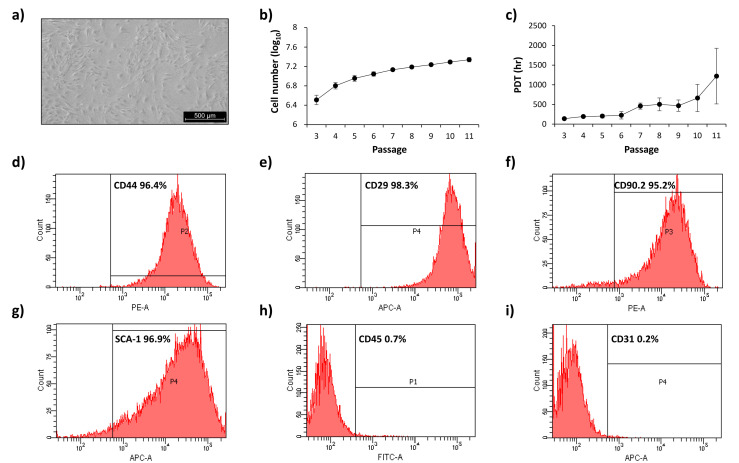
mADSCs characterization. (**a**) Representative photographs (5× magnification of an optical microscope, scale bar 500 µm) of mADSCs isolated from the subcutaneous adipose tissue of C57bl6 mice cultured in selection medium for 4 passages. (**b**) Accumulated cell number and (**c**) population doubling time (PTD) per hours of mADSCs from p3 to p11. Once 90% confluence was reached, cells were counted by an automated cell counter. Dots in line graphs represent the mean ± SEM values. (**d**–**i**) Representative histograms of the expression of CD44, CD29, CD90.2, Sca-1, CD45 and CD31 surface markers analyzed by FACS.

**Figure 2 cells-12-01741-f002:**
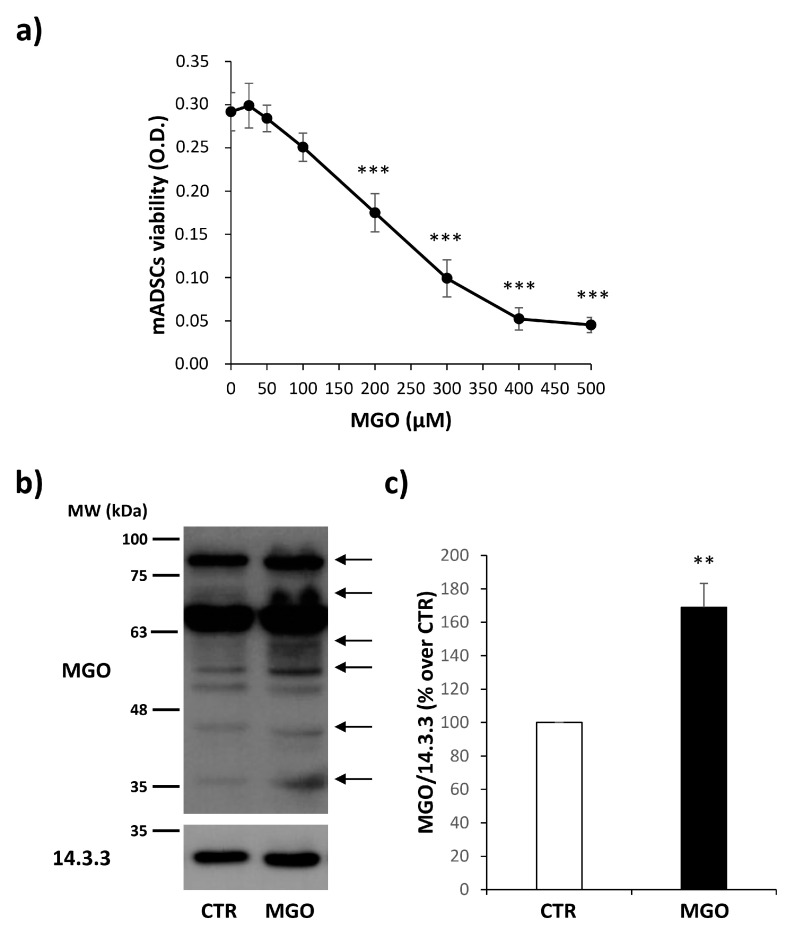
Dose–response curve and MGO-adduct accumulation in mADSCs. (**a**) mADSCs were exposed to increasing concentrations of MGO (25 μM, 50 μM, 100 μM, 200 μM, 300 μM, 400 μM and 500 μM) for 16 h. Data points show the mean values ± SEM of the optical density (O.D.) of 8 independent experiments recorded during the MTT assay. (**b**) Intracellular formation of MGO-modified proteins (MGO-adducts) analyzed by Western Blot on protein lysates obtained from mADSCs treated or not treated with MGO 100 μM. Bands, indicated by the arrows on the right side of the blot, show a spectrum of proteins modified by MGO. (**c**) The graph shows the densitometric analysis of MGO adducts levels normalized on 14.3.3. Bars in the graph show the mean ± SEM of 4 independent experiments. Statistical significance was assessed using Student’s *t*-test (** *p* ≤ 0.01, *** *p* ≤ 0.001).

**Figure 3 cells-12-01741-f003:**
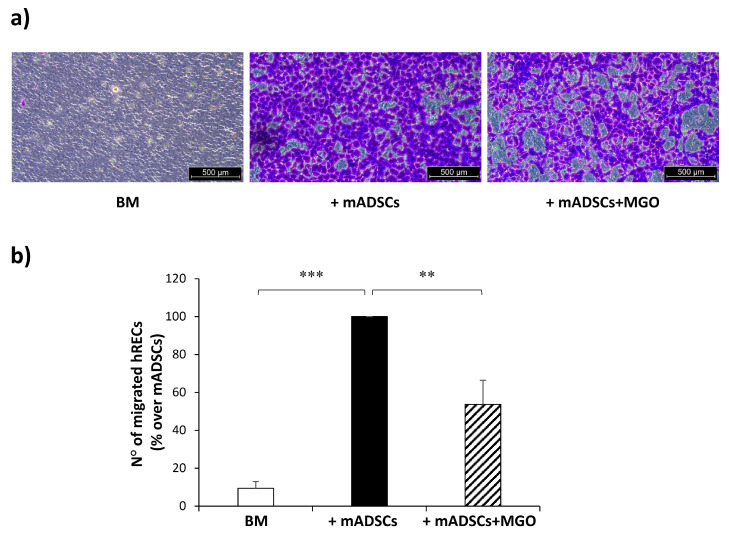
hREC migration in response to mADSC co-culture. (**a**) Representative photographs (5× magnification of an optical microscope, scale bar 500 µm) of migrated hRECs in the absence of mADSCs (BM, basal medium), in the presence of control mADSCs (+mADSCs) or MGO-treated mADSCs (+mADSCs + MGO). Migrated cells were stained with crystal violet and are visible in a violet color. (**b**) Bars in the graph represent the mean ± SEM of migrated hREC number of 8 independent experiments, expressed as % over mADSCs. Statistical significance was assessed using Student’s *t*-test (** *p* ≤ 0.01, *** *p* ≤ 0.001).

**Figure 4 cells-12-01741-f004:**
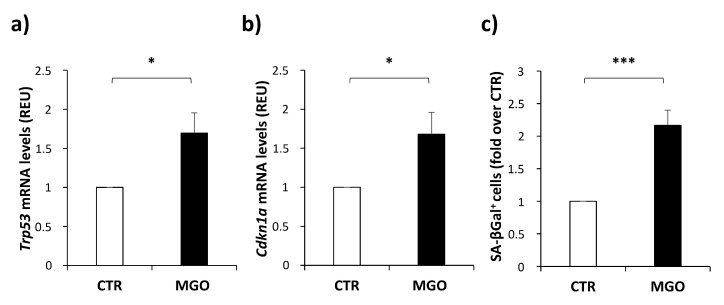
Senescence markers in mADSCs treated or not treated with MGO. mRNA levels of *Trp53* (**a**) and *Cdkn1a* (**b**) were measured by qPCR and normalized to *Cyclophilin A* expression in mADSCs, treated or not treated with MGO. (**c**) Flow cytometric detection of SA-βGal^+^ cells in mADSCs treated (MGO) or not treated (CTR) with MGO. Graphs show the mean ± SEM of 5 independent experiments. Statistical significance was assessed using Student’s *t*-test (* *p* ≤ 0.05; *** *p* ≤ 0.001).

**Figure 5 cells-12-01741-f005:**
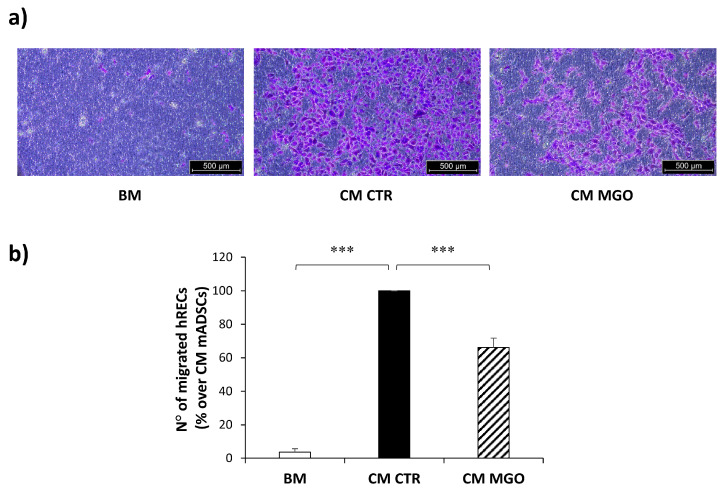
hREC migration in response to CM from mADSCs. (**a**) Representative photographs (5× magnification of an optical microscope, scale bar 500 µm) of migrated hRECs in response to basal medium (BM), CM from untreated mADSCs (CM CTR) or CM from MGO-treated mADSCs (CM MGO). Cells migrated to the lower side of the insert are visible in violet. (**b**) Bars in the graph represent the mean ± SEM of migrated hREC number in 8 independent experiments, expressed as % over CM mADSCs. Statistical significance was assessed using Student’s *t*-test (*** *p* ≤ 0.001).

**Figure 6 cells-12-01741-f006:**
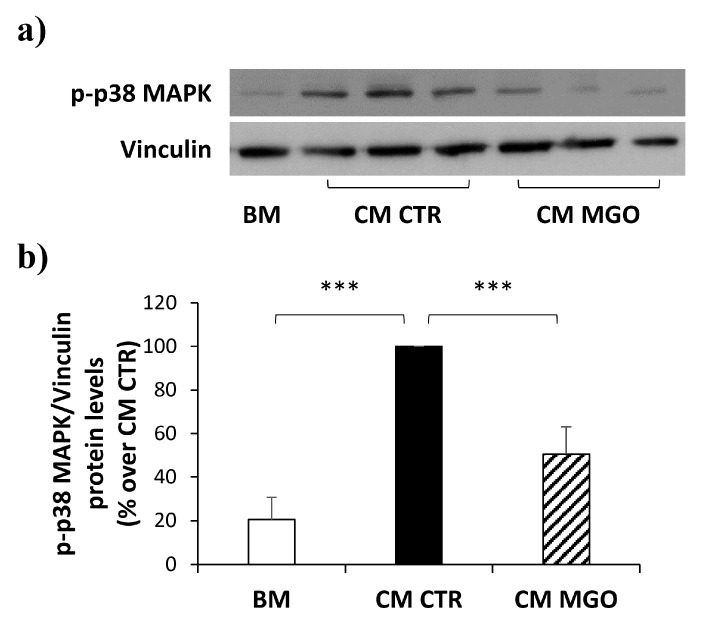
Phospho-p38 MAPK protein levels in hRECs exposed to mADSCs-CM. (**a**) The levels of p-p38 MAPK were analyzed by Western Blot on protein lysates obtained from hRECs exposed to CM from mADSCs treated or not treated with MGO; (**b**) the graph shows the densitometric analysis of p-p38 MAPK levels normalized on Vinculin. Bars in the graph show the mean ± SEM of 3 independent experiments. Statistical significance was assessed using Student’s *t*-test (*** *p* ≤ 0.001).

**Figure 7 cells-12-01741-f007:**
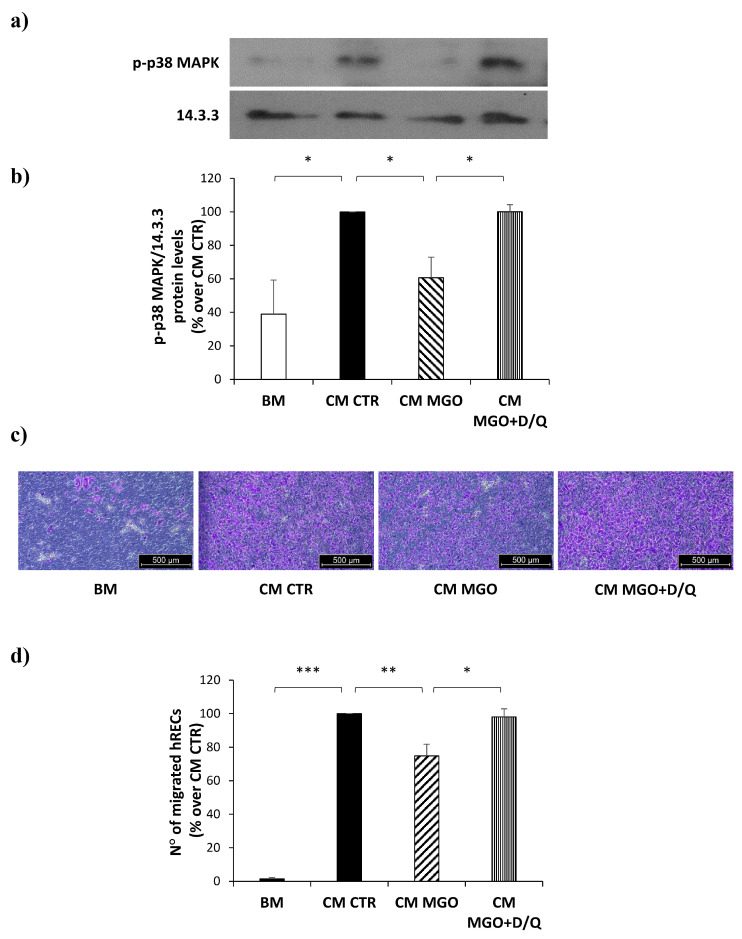
The effect of senolytic treatment on phospho-p38 MAPK protein levels and migration ability in hRECs exposed to CM from mADSCs. (**a**) Levels of p-p38 MAPK were analyzed by Western Blot on protein lysates obtained from hRECs exposed to CM from mADSCs treated (CM MGO) or not treated (CM CTR) with MGO, and then exposed to dasatinib (0.25 µM) and quercetin (10 µM) in combination (CM MGO + D/Q) for 24 h. Basal medium (BM) was used as negative control. (**b**) Bars in the graph show the mean ± SEM of 3 independent experiments, expressed as % over CM CTR, of the densitometric analysis of p-p38 MAPK levels normalized on 14.3.3. (**c**) Representative photographs (5× magnification of an optical microscope, scale bar 500 µm) of migrated hRECs in response to basal medium (BM), CM from untreated mADSCs (CM CTR), CM from MGO-treated mADSCs (CM MGO) or CM from mADSCs treated with MGO and senolytics (CM MGO + D/Q). Cells migrated to the lower side of the insert are visible in violet. (**d**) Bars in the graph show the mean ± SEM of 3 independent experiments, expressed as % over CM CTR, of the number of migrated hRECs. Statistical significance was assessed using Student’s *t*-test (* *p* ≤ 0.05, ** *p* ≤ 0.01, *** *p* ≤ 0.001).

**Table 1 cells-12-01741-t001:** List of primers used in this study.

Gene Name	Primer Sequence (5′ to 3′)
*Cyclophilin A:*	Forward GCAGACAAAGTTCCAAAGACAG Reverse CACCCTGGCACATGAATCC
*Fatty acid binding protein 4* *(Fabp4 or Ap2):*	Forward TCTCACCTGGAAGACAGCTCC Reverse GCTGATGATCATGTTGGGCTTGG
*Peroxisome proliferator-activated* *receptor gamma 2 (Pparγ2):*	Forward CAGTGGAGACCGCCCAGGCT Reverse TGGAGCAGGGGGTGAAGGCT
*Glucose transporter member 4 (Glut4):*	Forward CAATGTCTTGGCCGTGTTGG Reverse GCCCTGATGTTAGCCCTGAG
*Adiponectin (AdipoQ):*	Forward CTGACGACACCAAAAGGGCTC Reverse TCCAACCTGCACAAGTTCCC
*Transformation-related protein 53 (Trp53):*	Forward CCTCTCCCCCGCAAAAGAAA Reverse GACTCCTCTGTAGCATGGGC
*Cyclin-dependent kinase inhibitor 1A (Cdkn1a):*	Forward GCAGACCAGCCTGACAGATTT Reverse CTGACCCACAGCAGAAGAGG
*Runt-related transcription factor 2 (Runx2):*	Forward AGTCCATGCAGGAATATTTAAGGC Reverse CCAAAAGAAGCTTTGCTGACA
*Msh homeobox 2 (Msx2):*	Forward CCAGACATATGAGCCCCACC Reverse ACAGGTACTGTTTCTGGCGG
*Osteopontin (Ocp):*	Forward: CCGAGGTGATAGCTTGGCTT Reverse: ACAGGGATGACATCGAGGGA
*Osteocalcin (Ocn* *):*	Forward: GGTAGTGAACAGACTCCGGC Reverse: GGGCAGCACAGGTCCTAAAT

**Table 2 cells-12-01741-t002:** Mouse cytokine, chemokine and growth factors released by mADSCs. Soluble factors in CM by mADSCs treated (MGO) or not treated (CTR) with MGO were measured by multiplex assay. Basal medium (Opti-MEM) was used as blank and subtracted to sample values, which were then normalized on µg of cellular proteins. Results are shown as mean ± SEM of 5 independent experiments. Statistical analysis was performed by Student’s *t*-test (* *p* ≤ 0.05).

Variables	CTR	MGO
IL-6 (pg/mL × 1/µg of proteins)	0.20 ± 0.03	0.32 ± 0.04 *
MCP-1(MCAF) (pg/mL × 1/µg of proteins)	52.15 ± 4.40	80.29 ± 8.44 *
IL-12p40 (pg/mL × 1/µg of proteins)	0.24 ± 0.03	0.35 ± 0.04 *
IL-1β (pg/mL × 1/µg of proteins)	0.009 ± 0.003	0.01 ± 0.0005
IL-5 (pg/mL × 1/µg of proteins)	0.04 ± 0.01	0.036 ± 0.01
Eotaxin (pg/mL × 1/µg of proteins)	1.16 ± 0.29	1.80 ± 0.46
G-CSF (pg/mL × 1/µg of proteins)	0.12 ± 0.07	0.17 ± 0.08
GM-CSF (pg/mL × 1/µg of proteins)	0.12 ± 0.05	0.08 ± 0.04
IFN-ƴ (pg/mL × 1/µg of proteins)	0.03 ± 0.01	0.03 ± 0.01
KC (pg/mL × 1/µg of proteins)	3.40 ± 0.80	4.33 ± 0.81
MIP-1β (pg/mL × 1/µg of proteins)	0.19 ± 0.04	0.21 ± 0.05
RANTES (pg/mL × 1/µg of proteins)	0.32 ± 0.08	0.41 ± 0.18
TNF-α (pg/mL × 1/µg of proteins)	0.05 ± 0.003	0.04 ± 0.01

**Table 3 cells-12-01741-t003:** Pro-angiogenic factors released by mADSCs. Soluble factors in CM from mADSCs treated (MGO) or not treated (CTR) with MGO were measured by multiplex assay. Basal medium (Opti-MEM) was used as blank and subtracted to sample values, which were then normalized on µg of cellular proteins. Results are shown as mean ± SEM of 4 independent experiments. Statistical analysis was performed by Student’s *t*-test (* *p* ≤ 0.05, ** *p* ≤ 0.01).

Variables	CTR	MGO
VEGF (pg/mL × 1/µg of proteins)	30.88 ± 3.02	18.34 ± 0.67 **
PDGF-BB (pg/mL × 1/µg of proteins)	0.31 ± 0.05	0.05 ± 0.06 *
Basic FGF (pg/mL × 1/µg of proteins)	1.13 ± 0.87	0.83 ± 0.67
M-CSF (pg/mL × 1/µg of proteins)	4.90 ± 0.94	3.83 ± 0.43
MIP2 (pg/mL × 1/µg of proteins)	0.07 ± 0.02	0.04 ± 0.01

**Table 4 cells-12-01741-t004:** Pro-angiogenic factors released by mADSCs following senolytic treatment. Soluble factors in CM from untreated mADSCs (CTR), treated with MGO (MGO) and treated with MGO and 0.25 µM dasatinib and 10 µM quercetin, in combination, (MGO + D/Q) were measured by multiplex assay. Basal medium (Opti-MEM) was used as blank and subtracted to sample values, which were then normalized on µg of cellular proteins. Results are shown as mean ± SEM of 3 independent experiments. Statistical analysis was performed by Student’s *t*-test (* *p* ≤ 0.05, *** *p* ≤ 0.001 vs. CTR; ## *p* ≤ 0.01, ### *p* ≤ 0.001 vs. MGO).

Variables	CTR	MGO	MGO + D/Q
VEGF (pg/mL × 1/µg of proteins)	31.28 ± 1.59	16.27 ± 1.42 ***	31.90 ± 0.78 ###
PDGF-BB (pg/mL × 1/µg of proteins)	0.54 ± 0.12	0.18 ± 0.07 *	0.55 ± 0.05 ##

## Data Availability

Not applicable.

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
