# Peer review of "Methylglyoxal Impairs the Pro-Angiogenic Ability of Mouse Adipose-Derived Stem Cells (mADSCs) via a Senescence-Associated Mechanism"

_cells, 2023, doi:10.3390/cells12131741_

Round 1

Reviewer 1 Report

the authors investigated the impact of diabetes on the pro-angiogenic function of AD-MSC, by administration of methylglyoxal MGO, a metaboltie that accumulate on cells diring diabetes progression.

The authors use animals, bacause AD-MSC derived from mice and they not reported the etichical committee authorization. Without this, manuscript could not be pubblisched!   

  in the introduction, pag 2,line 59 I suggest to add a couple of sentences reporting some exmaples of literature. From line 85-91 pag2, I suggest to move in discussion seciton. 

Methods: 

pag 3, line 138, please add the details of fat differentiation protocol. What type of media did you use? 

about migration and co-colture test, pag 5 line 180-193 there are several issues. spcifcy the colture media used, since that Ad-MSC are cultured with DMEM/F12 supplemented with FBS and hREC with Endothelial Cell Medium. Is Optimem (lined 191) a starvation media? the starvation is a critical passage and you need to better clarify. Finaly, insert the crystal violet protocol, please.

Results: 

in paragraph 3.1 , please add the amount of fat used to isolate MSC and the amount of cells obtained after digestion and after amplification. Please report a population doubling cells and a picture of mAD-MSC culture. the authors conclude to characterize AD-MSC, Do you test other antibodies? such as stro-1, CD90? What are the criteria to establish that you obtained AD-MSC from mouse? For isolation of humans AD-MSC thare are well established criteria reported by Bourin et al.  https://doi.org/10.1016/j.jcyt.2013.02.006

Please, add in the Supplementary figure the bone differentiation. in pciture of oil red staining add in figure legend the lenght of scal bar. 

results about cytokines are obtained after starvation? or with culture media DMEM/F12 supplmented with FBS? FBS could contribute to quantification of cytockines,  I sugget to add a line "blank" with dosage of cytokines in culture media. 

Please introduce Figure 3, figure 5, figure 7 scale bar. 

Discussion:

lane 385-394, please investgate the characterization of mAD-MSC, considering the reduce commercially availability of antibodies specific for mouse. it's necessary to give the evidence of bone differntiation abiltiy of MSC. 

REFERNCES: thare are too many references for an original work, I suggest to reduce the number up to 40-45 refences. 

Author Response

the authors investigated the impact of diabetes on the pro-angiogenic function of AD-MSC, by administration of methylglyoxal MGO, a metaboltie that accumulate on cells diring diabetes progression.

Point 1. The authors use animals, bacause AD-MSC derived from mice and they not reported the etichical committee authorization. Without this, manuscript could not be pubblisched!  

Response1:  We thank the Reviewer for his comment. Tissue collection from mice was approved by the local ethics committee of the Ministry of Health (approval n. 252/218-PR). The latter is now included in the revised version of the manuscript (P3 L120,121 and P16 L772-774)

Point 2.   in the introduction, pag 2,line 59 I suggest to add a couple of sentences reporting some exmaples of literature. From line 85-91 pag2, I suggest to move in discussion seciton.

Response 2: A couple of references about the issue have been included in the new introduction section, as requested (P2 L65-70).

The main conclusions of the work were highlighted at the end of the introduction section as indicated in the authors’ guidelines. These conclusions have been already added following the Discussion section (P16 L749-758). In case the Reviewer requests lines 85-91 to be removed from the introduction, we will ask to the editorial office.

Methods:

Point 3. pag 3, line 138, please add the details of fat differentiation protocol. What type of media did you use?

Response 3: A more detailed description of the adipocyte medium used in the differentiation protocol has been included in the new version of the manuscript (P4 L164, 165).

Point 4. about migration and co-colture test, pag 5 line 180-193 there are several issues. spcifcy the colture media used, since that Ad-MSC are cultured with DMEM/F12 supplemented with FBS and hREC with Endothelial Cell Medium. Is Optimem (lined 191) a starvation media? the starvation is a critical passage and you need to better clarify. Finaly, insert the crystal violet protocol, please.

Response 4: We thank the Reviewer for this comment. We have now included missing information about culture media and crystal violet protocol in the new version of the manuscript (P6 L251-263 and P6 L265-270).

Opti-MEM does not contain serum and, as such, it can be defined as a starvation medium (doi:10.1152/ajpcell.00091.2011).     

Results:

Point 5. in paragraph 3.1 , please add the amount of fat used to isolate MSC and the amount of cells obtained after digestion and after amplification. Please report a population doubling cells and a picture of mAD-MSC culture. the authors conclude to characterize AD-MSC, Do you test other antibodies? such as stro-1, CD90? What are the criteria to establish that you obtained AD-MSC from mouse? For isolation of humans AD-MSC thare are well established criteria reported by Bourin et al.  https://doi.org/10.1016/j.jcyt.2013.02.006

Response 5: The amount of fat and cells obtained after digestion and amplification is included in paragraph 3.1 (P7 L308-311) and new figure 1b, as requested.

Population doubling time and a picture of mADSC are included in new figure 1a and 1c.

We characterized the ADSC by the three minimum criteria established by the ISCT guidelines (adhesion growth, immunophenotype and ability to differentiate in other cell lines). For osteogenic differentiation, please look at response to point 6. The immunophenotype specific for mouse AD-MSCs has been delineated by several authors in literature [doi: 10.1186/s12575-019-0091-3; doi: 10.1016/j.transproceed.2008.08.009; doi: 10.1016/j.tice.2010.04.003; doi: 10.3727/096368913x664586]. To meet the reviewer’s request, in line with these publications, we have expanded the mADSC characterization described in figure 1 with the surface expression markers CD90 (new figure 1f) and Sca-1 (new figure 1g). Please look at new figure 1 and Results (P7 L315, 316), Materials and Methods (P3 L114 and 139-142) and Discussion (P14 L594-596) sections of the revised version of the manuscript.

Although Stro-1 is used as marker of human MSCs, it has been described to be not expressed on mouse MSCs [doi:10.3727/096368912x655127; doi:10.1186/ar2116]. 

Point 6. Please, add in the Supplementary figure the bone differentiation. in pciture of oil red staining add in figure legend the lenght of scal bar.

Response 6: New data showing osteoblastic differentiation of mADSC are reported in Supplementary Figure 2 and described in the Materials and Methods, Results and Discussion sections [P4 L177-193; P7 L319-322; P14 L598] of the revised version of the manuscript. Scale bar length has been included in the figure legend of Supplementary Figure 1.

Point 7. results about cytokines are obtained after starvation? or with culture media DMEM/F12 supplmented with FBS? FBS could contribute to quantification of cytockines,  I suggest to add a line "blank" with dosage of cytokines in culture media.

Response 7: We thank the Reviewer for this point. Cytokines were measured in conditioned media, obtained after 24h of cell culturing in starvation medium (Opti-MEM simple medium with no serum). Please, also look at response to point 4.

In multiplex cytokine assay, starvation medium was used as blank and, as such, subtracted to sample values, which are shown as normalized on cellular proteins. Because the normalization on cellular proteins is required for sample values but not for blank values, the latter are not listed in tables but subtracted to sample measurements. We clarified this issue in the Materials and Methods section and legends of tables 1, 2 and 3 (P6 L282, 283; P10 L486, 487; P11 L515, 516; P12 L546) of the revised version of the manuscript.

Point 8. Please introduce Figure 3, figure 5, figure 7 scale bar.

Response 8: Done as requested.

Discussion:

Point 9. lane 385-394, please investgate the characterization of mAD-MSC, considering the reduce commercially availability of antibodies specific for mouse. it's necessary to give the evidence of bone differntiation abiltiy of MSC.

Response 9: This point has been addressed at lines 594-598, as requested.

Point 10. REFERNCES: thare are too many references for an original work, I suggest to reduce the number up to 40-45 refences.

Response 10: To meet the Reviewer’s request, 19 citations have been removed from the original reference list.

Reviewer 2 Report

In the present study, authors evaluate the effect of Methylglyoxal (MGO), a product of chronic 14 hyperglycemia, on mouse ADSCs (mADSCs) pro-angiogenic function and the molecular mediators involved.
 ADSCs were isolated from subcutaneous adipose tissue biopsies of 4 months old C57bl6 mice. How many cell strains were used in the present study?

Fig 2b, Western blot Bands, indicated by the arrows on the right side of the blot, show a spectrum of proteins modified by MGO. The proteins modified by MGO should be identified.

How about the therapeutic effects of MGO-treated ADSCs. In vivo experiment should be done.

Author Response

In the present study, authors evaluate the effect of Methylglyoxal (MGO), a product of chronic 14 hyperglycemia, on mouse ADSCs (mADSCs) pro-angiogenic function and the molecular mediators involved.

Point 1. ADSCs were isolated from subcutaneous adipose tissue biopsies of 4 months old C57bl6 mice. How many cell strains were used in the present study?

Response 1: Data collected in this study were obtained from at least 3 different isolation of mADSC from C57bl6 mice. This information is now included in the Materials and Methods section of the new version of the manuscript (P3 L131, 132).

Point 2. Fig 2b, Western blot Bands, indicated by the arrows on the right side of the blot, show a spectrum of proteins modified by MGO. The proteins modified by MGO should be identified.

Response 2:  The purpose of the experiment in figure 2b is to show that our experimental condition is enough to increase the amount of proteins modified by MGO (MGO-adducts) [see also doi.org/10.7554/eLife.58573], as physiologically happens in response to MGO accumulation, typically in diabetes and aging. Dedicated studies have been performed by others to identify the so called “dicarbonyl proteome”, including hundreds of proteins susceptible to MGO modifications, by proteomic approaches [doi.org/10.3390/ijms23073689; doi.org/10.1016/j.bbrc.2015.01.140]. Therefore, ad hoc studies requiring MS analysis would be needed, but this is beyond the scope of the present work. We have better clarified this point in the new version of the manuscript (P14 L605-613).

Point 3. How about the therapeutic effects of MGO-treated ADSCs. In vivo experiment should be done.

Response 3: In our work we have shown that MGO treatment impairs ADSC pro-angiogenic function by inducing senescence. It is already known that senescent ADSCs loose their characteristics and angiogenic potential [doi.org/10.1186/s13293-019-0263-5; doi.org/10.3390/ijms21051802; doi.org/10.1186/s13287-021-02509-0; doi.org/10.3389/fcell.2020.00364], and transplantation of senescent mADSC causes physical dysfunction in mice [doi.org/10.1038/s41591-018-0092-9]. Also, we recently showed that MGO induces cellular senescence in vivo [doi.org/10.15252/embr.202152990]. We have better discussed this important issue in the new version of the manuscript (P15 L702-704).

Reviewer 3 Report

The study was well designed and adequately conducted: the results are very interesting 

Author Response

Point 1. The study was well designed and adequately conducted: the results are very interesting

Response 1: We thank the Reviewer for his comment.

Reviewer 4 Report

In this study, the authors reported that MGO accumulation impairs the pro-angiogenic function of mADSCs, shown as the impaired mADSCs ability to release both VEGF and PDGF, and induce hRECs migration. They point at the mADSCs senescence as the underlying mechanism that contributes to the reduced activation of p38-MAPK. This study provides novel information about the direct harmful effect of MGO on mADSCs functionality, paving the way for the optimization of therapeutic strategies involving the use of autologous ADSCs for the treatment of diabetic-associated vascular defects. The study is designed appropriately. The results are clearly presented and supported the conclusions. I recommend it can be accepted for publication after minor revise. In the Figure 1, “0,7%”, “98,3”, “0,2”, and “96,4” should be revised as “0.7%”, “98.3”, “0.2”, and “96.4”.

Author Response

In this study, the authors reported that MGO accumulation impairs the pro-angiogenic function of mADSCs, shown as the impaired mADSCs ability to release both VEGF and PDGF, and induce hRECs migration. They point at the mADSCs senescence as the underlying mechanism that contributes to the reduced activation of p38-MAPK. This study provides novel information about the direct harmful effect of MGO on mADSCs functionality, paving the way for the optimization of therapeutic strategies involving the use of autologous ADSCs for the treatment of diabetic-associated vascular defects. The study is designed appropriately. The results are clearly presented and supported the conclusions. I recommend it can be accepted for publication after minor revise.

Point1. In the Figure 1, “0,7%”, “98,3”, “0,2”, and “96,4” should be revised as “0.7%”, “98.3”, “0.2”, and “96.4”.

Response 1: We thank the Reviewer for his comments. Figure 1 has been revised as requested, and replaced in the new version of the manuscript.  

Round 2

Reviewer 1 Report

The authors have implemnted the manuscript with recommended information/revision and from my side is accepted for pubblication